# Efficacy of L-Arabinose in Lowering Glycemic and Insulinemic Responses: The Modifying Effect of Starch and Fat

**DOI:** 10.3390/foods11020157

**Published:** 2022-01-08

**Authors:** Korrie Pol, Marie-Luise Puhlmann, Monica Mars

**Affiliations:** 1Division of Human Nutrition and Health, Wageningen University, P.O. Box 17, NL-6700 AA Wageningen, The Netherlands; korrie.pol@wur.nl (K.P.); marie-luise.puhlmann@wur.nl (M.-L.P.); 2Lab of Microbiology, Wageningen University, P.O. Box 17, NL-6700 AA Wageningen, The Netherlands

**Keywords:** L-arabinose, sucrose, fat, starch, glucose, insulin, GLP-1, GIP, healthy adults

## Abstract

L-arabinose is a bio-active compound derived from the side-streams of plant food processing. L-arabinose lowers glycemic and insulinemic responses when added to simple water-based sugary liquids. However, the effect in more complex foods, including fat and starch, is inconsistent. This study assessed the effect of fat or starch in a sugary drink on the efficacy of L-arabinose. Twenty-three healthy volunteers (12 female/11 male; aged 24 ± 3 years; BMI 23 ± 3 kg/m^2^) participated in a randomised cross-over trial with six drinks: control: 50 g sucrose in water; fat: control + 22 g oil; starch: control + 50 g starch; and all three with and without the addition of 5 g L-arabinose. The addition of L-arabinose to the control drink lowered glucose and insulin peaks by 15% and 52%; for the fat drink by 8% and 45%; and for the starch drink by 7% and 29%. For all three drinks, adding L-arabinose increased glucagon-like peptide 1 (GLP-1) responses and lowered Glucose-dependent insulinotropic polypeptide (GIP) responses. Despite adding large quantities of starch and fat to sugary drinks, L-arabinose significantly lowered postprandial glycemic and insulinemic responses in healthy subjects. These findings suggest that L-arabinose can be functional in more complex foods; however, the factors affecting its efficacy in solid food matrices need to be studied in more detail.

## 1. Introduction

Worldwide, more than 463 million people are currently diagnosed with diabetes and 374 million with impaired glucose tolerance [1]. Postprandial glycemic control is of high importance in these populations. Systematic reviews have shown that diets containing mainly foods with a low potency to increase blood glucose levels, so-called low glycemic index (GI) diets, improve biomarkers of glycemic control [2,3]. In addition, several clinical guidelines advise diabetic patients to especially incorporate foods with a low GI into their diets [4,5,6].

Simple sugars, such as sucrose, are often added to foods for their sweet taste. However, they are also known for their fast and high postprandial glycemic responses. The food industry has a high interest in novel functional ingredients that can lower the glycemic potency of sugar-rich foods without compromising the food’s sweet taste and other food properties such as texture. L-arabinose is such a functional ingredient. L-arabinose is an aldopentose molecule with a relatively sweet taste [7]. L-arabinose can be derived from hemicellulose by enzymatic hydrolysis [8]. Hemicellulose is abundantly present in the side-streams of plant food processing, such as sugar production [8]. L-arabinose has been shown to slow down glucose absorption by inhibiting the intestinal enzyme sucrase in an uncompetitive manner [9]. Sucrase is present in the brush border of the intestine and critical for hydrolysing sucrose into a glucose and fructose molecule [9].

Several randomised clinical trials with healthy subjects show that adding L-arabinose to simple drinks or gels containing sucrose reduces postprandial glycemic responses significantly [10,11,12]. When 0.9 g L-arabinose was added to a tropical fruit-flavoured drink sweetened with 10 g sucrose, glycemic and insulinemic responses were lowered by 9% and 35%, respectively [13]. In addition, the replacement of 30% sucrose by L-arabinose lowered glycemic and insulinemic responses by 22% and 67% [14]. However, the efficacy of L-arabinose is less evident for more complex foods. For example, no effect was found on glucose or insulin responses when L-arabinose was incorporated into a mixed breakfast meal [15], cereals [14], and when sucrose was replaced with L-arabinose in muffins [13]. Although in the latter two studies no effect was found on glycemic response, trends towards lower insulinemic responses were observed [13,14]. These findings suggest that L-arabinose may have had an inhibiting effect on sucrose breakdown, but that the net effects could not be observed in the circulating glucose levels of healthy subjects. In addition, it may very well be that the other components present in the foods, such as fat and starch, may have modified the effect. For example, fat is known to slow down gastric emptying, and may therefore also slow down glucose absorption [16,17]. At the same time, the abundance of starch present in the foods may overshadow the effect of an inhibited sucrose breakdown. Therefore, the impact that individual food components can have on the efficacy of L-arabinose needs to be disentangled before L-arabinose can be implemented into real life and complex foods.

This manuscript describes the results of a model food experiment that was conducted in order to explore the modifying effect of isolated food components on the efficacy of L-arabinose in sucrose-rich foods. The model foods were sucrose drinks containing large quantities of either starch or fat with or without the addition of L-arabinose. With this model food experiment, the opportunities and bandwidth of food applications are illustrated. It is demonstrated that L-arabinose still lowers the postprandial glycemic and insulinemic response in healthy subjects, despite the presence of starch and fat. Moreover, effects are shown on associated physiological markers—that is, the incretin hormones glucagon-like peptide 1 (GLP-1) and glucose-dependent insulinotropic polypeptide (GIP). These findings suggest that L-arabinose maintains its functionality in the presence of starch and fat, two important compounds present in more complex foods and real-life foods.

## 2. Materials and Methods

### 2.1. Subjects

Healthy adults (both sexes, age 18–35 years) were recruited in Wageningen and surrounding areas. For the recruitment, emails were sent to persons in a database of volunteers who had previously expressed their interest in participating in nutrition studies. Inclusion criteria were as follows: apparently healthy, as judged by the subject and measured by a health and lifestyle questionnaire (i.e., no known diseases, no medicine use, and no allergy or intolerance to the food products under study); normal fasting glucose (<6.1 mmol/L); normal haemoglobin concentration (>8.5 mmol/L for males and >7.5 mmol/L for females); and able to communicate in spoken and written Dutch. Subjects who were pregnant or lactating or reported weight fluctuations of >5 kg in the two months prior to the screening or consumed excessive alcoholic beverages (≥21 glasses/week) were excluded.

A total of 87 subjects completed the health and lifestyle questionnaire to determine their eligibility. In addition, the Dutch Eating Behaviour Questionnaire (DEBQ) was completed to identify eating restraint, emotional eaters, and external eaters [18]. However, none of these personality traits affected the glycemic or insulinemic responses, hence they will not be further discussed. Eligible subjects came after fasting to a physical examination, during which weight and height were measured, blood glucose and haemoglobin were measured with a finger prick, and a nurse checked the suitability of the participants’ antecubital veins for inserting a cannula several times.

In total, 25 subjects were included in the trial, of which 23 subjects completed all treatments (Table 1). Two subjects dropped out: one of them was not able to consume the total amount of the product (Fat + Ara), while the other subject dropped out due to side effects of the blood drawings (painful elbow cavities due to a combination of the cannula and playing volleyball). The data of these 2 subjects were not analysed.

Subjects received financial compensation after participation. The study was performed in accordance with Dutch law and the principles laid down in the current version of the Declaration of Helsinki. The medical ethical committee of Wageningen University gave ethical approval (ABR nr: NL61428.081.17). All eligible subjects provided oral and written informed consent. The trial and its primary outcomes were pre-registered in the Dutch Trial Register as NTR6636 (www.trialregister.nl Trial NL6458).

The sample size was based on the effect of L-arabinose on glycemic response peaks. Sample size estimations were performed with the software program G-power (University of Kiel, Kiel, Germany). For all calculations, a paired Student’s *t*-test was assumed, with a two-tailed *t*-test with α = 0.05, power = 0.8, and SD of 0.75 mmol/L (obtained from previous experiments [13,14]). Furthermore, an effect of 0.5 mmol/L was assumed to be relevant in this population, which means an effect size (dz) of 0.67. This resulted in an estimated sample size of minimal 20 participants. Previous studies from our lab [13,14] showed dropout rates of 17% and 11%. Taking into account a safe dropout rate of 20%, it was decided to enrol 24 subjects.

### 2.2. Design

The study was a 3 × 2 factorial randomised cross-over trial. The study was carried out between August and October 2017 at the division of Human Nutrition and Health, Wageningen University, The Netherlands. The treatments consisted of three drinks (control drink, fat drink, or starch drink) with or without 0 g or 5 g L-arabinose (see Section 2.3 Test Products).

The researchers and subjects were blinded for the addition of L-arabinose, but not for the type of drink (control, fat, starch), due to the difference in mouthfeel and taste. Subjects were randomised to all six treatments. An independent research assistant generated a list with randomised sequences, using a Williams design. Another independent research assistant was responsible for the allocation of the randomisation codes to the products. The researchers allocated the subjects in consecutive order to the randomized sequences. To minimise carry-over effects, a washout period of a minimum of three days was used.

### 2.3. Test Products

Ingredients, energy content, and liking of the test products are described in Table 2. All test drinks contained 50 g of sucrose and 500 g of water. The L-arabinose-containing drinks contained 5 g of L-arabinose (10% *w*/*w* sucrose). The preparation protocol of the drinks was based on extensive in vitro experimenting (see Appendix A). The drinks were freshly prepared every morning according to the preparation protocol and were distributed in identical opaque cups wrapped in aluminium foil. The subjects received the following instructions: “Drink the full amount gradually within 5 min with a straw”. As the test products were clearly different in taste and mouthfeel, it was only possible to blind the subjects to the presence or absence of L-arabinose, but not to the presence of starch or fat.

The subjects scored the drinks on a liking scale directly after they finished the drinks. A 7-point Likert scale ranging from “not liked at all” to “very much liked” was used. On average, the control drinks were scored as 5.1 ± 1.3, the fat drinks as 4.2 ± 1.2, and the starch drinks as 2.9 ± 1.4 (Table 2). The addition of L-arabinose did not change the liking score of the drinks (all *p*-values > 0.05).

#### 2.3.1. Test Drink Preparation

##### Control Drinks

The control drinks consisted of sugar solutions with 9% (*w*/*w*) sucrose, and in the case of Control + Ara, also 0.9% (*w*/*w*) L-arabinose (Cosun, Breda, The Netherlands). Sugar (and L-arabinose) were mixed with cold tap water in a Waring blender twice for 10 s (CB102T, Waring, Torrington, CT, USA). These sugar solutions were then also used as a base for the fat and starch drinks.

##### Fat Drinks

The fat drinks were a 3.8% (*w*/*w*) oil-in-water emulsion using sunflower oil and 0.4% (*w*/*w*) Tween 80 (Lamesorb SMO 20, BASF, Basel, Germany). The emulsifier was added to prevent the fat from phase separating beforehand or in the stomach. The sunflower oil and the emulsifier were blended into the pre-made sugar solutions using a Waring blender for 90 s at full speed (HGB500, Waring, Torrington, CT, USA). Both fat treatments had the same average droplet sizes, which were d3,2 = 1.63 µm and d4,3 = 4.54 µm (Appendix A).

##### Starch Drinks

The starch drinks were a dispersion of a pregelatinised waxy maize starch (C Gel–Instant 12410, Cargill, Minneapolis, MN, USA) that can be solubilised in water without heating. The starch drinks were prepared by quickly and carefully incorporating the pregelatinised starch into the premade sugar solutions using a hand blender set to full speed for at least 30 s (MSM87110, Bosch, BSH group, Nazarje, Slovenia).

### 2.4. Study Procedures

#### 2.4.1. Before the Test Day

To standardise the fasting state, subjects were asked to maintain their habitual diet and other lifestyle habits throughout the whole study period. Subjects were not allowed to drink alcohol nor to perform heavy exercise two days before test days. Subjects reported their evening meal before the first test session in their study diary—that is, the ingredients, preparation methods, and the amounts they ate in household measures. Additionally, subjects registered the amount of physical activity they did the evening before the test day. They were then instructed to eat and exercise in the same way in the evenings prior to the test day.

#### 2.4.2. Test Days

The subjects arrived at the study site between 7.30 and 8.00 a.m. after an overnight fast; subjects did not eat or drink anything other than water, coffee, or tea between 20:00 and 22:00 h and were only allowed to drink water from 22:00 h the evening before the test. The study personnel checked the diaries for deviations from the study protocol and the subjects then completed the well-being and gastro-intestinal comfort questionnaire. Then, a trained and experienced nurse inserted an intravenous cannula. After at least 30 min rest, the nurse drew the baseline blood samples and the subjects completed an appetite questionnaire. After that (*t* = 0 min), the subjects drank the test product within 5 min. After the drink was consumed, subjects drank an additional 135 mL of water and scored the test product on liking (see Section 2.3 Test Products). Blood samples and appetite questionnaires were conducted at *t* = 15, 30, 45, 60, 90, 120, 180 min after the subjects started to consume the test products. After 120 min, subjects again drank 135 mL of water. At *t* = 180 min, a gastro-intestinal comfort questionnaire was completed. Subjects remained seated in the living room of the study facility and could do sedentary activities such as quietly reading, watching television, or listening to music. After the last blood sample was drawn, the intravenous cannula was removed. Subjects were then directed to the eating lab and received an *ad libitum* meal (see Section 2.6). The morning after the test day, the subjects received an online evaluation questionnaire and were asked which experimental product they thought they had had. Subjects were instructed to fill out a study diary throughout the whole study period in which they could report any complaints and deviations from the study protocol. In addition, a well-being questionnaire was filled out to register any adverse effects(see Section 2.7 Gastro-Intestinal Tolerance).

### 2.5. Biochemical Measures

Venous blood samples for plasma glucose analysis were collected into NaF vacutainers (Becton Dickinson, Franklin Lakes, NJ, USA), and for insulin analyses, blood samples were collected into EDTA vacutainers (Becton Dickinson, Franklin Lakes, NJ, USA) and kept in ice water for a maximum of 15 min before being centrifuged. For GLP-1 and GIP analyses, EDTA + Aprotinine vacutainers were manually pre-treated by injecting 50 μL DPP-IV inhibitor (Catalog no. DPP4-010, Millipore, Burlington, MA, United States USA) to prevent proteolytic cleavage. These tubes were kept in ice water before use and venous blood samples were collected into these vacutainers and put in ice water for a maximum of 15 min before being centrifuged. All tubes were centrifuged for 10 min at 1200× *g* at 4 °C. Plasma was then aliquoted and stored at −80 °C until analysis.

Plasma glucose samples were measured with the hexokinase assay using the Siemens Dimension Vista System (Siemens Healthcare, The Hague, The Netherlands). Plasma insulin samples were measured using an enzyme-linked immunosorbent assay (ELISA) (catalogue no. 10-1113-10, Mercodia Insulin ELISA, Uppsala, Sweden) (the lowest detectable level for insulin was 1.0 mU/L; intra-assay-coefficient of variation (CV): 4% and inter-assay CV: 4%). Plasma total GLP-1 and GIP were measured with a custom-made multiplex assay (Meso Scale Discovery, Rockville, MD, USA). All samples of one subject were analysed within one run or plate, which also contained a positive and negative control.

### 2.6. Subjective Appetite Ratings and Ad Libitum Food Intake

Hunger, fullness, desire to eat, prospective food consumption, and thirst were measured with 100 mm visual analogue scales (VAS) with the anchors: “not at all” (left) to “very much” (right) and filled out at the same time points as the blood drawings [19].

An *ad libitum meal* was consumed after 3 h. Subjects received 18 small wheat buns (~22 g each) and several toppings, being 100 g of low-fat margarine, 8 slices of cheese, 100 g of strawberry jam, and 100 g of chocolate sprinkles. Further, 500 mL water, 500 mL hot water with tea bags, 500 mL coffee, and sachets of creamer powder and sugar were offered. Subjects were instructed to eat as much as they wanted until they felt comfortably full. Left-overs were covertly weighed. Food intake was calculated by subtracting the weight of the leftovers from the food that was provided. Energy and macronutrient intake was estimated using the Dutch Food Composition Database [20]. None of the subjects finished all buns or asked for a second portion.

### 2.7. Gastro-Intestinal Tolerance

Potential impact on gastro-intestinal tolerance was measured with questions on bloating, regurgitation, flatulence, and nausea, giving a grade being no (0), little (1), moderate (2), or severe (3) at the study site at *t* = 0 and *t* = 180 min. Further, similar questions were completed three times in the diary: on days preceding a study day, on every study day, and the day after a study day.

### 2.8. Calculations and Statistical Analysis

Several parameters were extracted from the individual time curves. For glucose, insulin, GLP-1, and GIP responses these were: the incremental area under the curve above the baseline (iAUC_0–180min_) by the trapezoidal rule, peak-value (C_max_), and time-to-peak (T_peak_). In addition, the area under the curve (AUC) for the appetite ratings (AUC_0–180min_) was calculated by the trapezoidal rule.

Outcomes are shown as least square means and 95% confidence intervals, unless stated otherwise. Statistical analyses were performed two-sided, and *p*-values < 0.05 were considered statistically significant. All data were analysed using SAS for Windows software v9.4 (SAS Institute Inc., Cary, NC, USA). All continuous variables were checked for normal distribution by visual inspection of histograms and QQ-plots of the Studentised residuals. There were ten missing values for GLP-1 and these were left missing in the mixed model analyses.

All statistical analyses were performed separately for control, fat, and starch drinks. Comparisons were made between arabinose and non-arabinose containing drinks. The responses over time (glucose, insulin, GLP-1, GIP, appetite score) were analysed using a linear mixed model ANOVA (PROC MIXED; SAS Institute Inc., Cary, NC, USA). The fixed factors were treatment (with or without L-arabinose), time, and the interaction of treatment x time, with time being a repeated factor and ID number as a random factor. The compound symmetry covariance structure was used, as this was the best-fitting covariance structure based on the lowest AIC. The contrasts between time points were assessed by treatment x time effects sliced by time. Post hoc tests were performed with Tukey correction. These results are presented in Figure 1. Moreover, iAUC_0–180min_ (total response), C_max_ (peak), and T_peak_ (time-to-peak) were compared between treatments with and without L-arabinose by means of a paired *t*-test. Mean ad libitum energy intake and the change in gastro-intestinal tolerance was compared between treatments with the mixed model ANOVA described above.

Before the actual statistical analyses, all endpoints were tested in a separate model to check for order and baseline effects; in these analyses, no statistically significant effects were found.

## 3. Results

### 3.1. Control Drink

#### 3.1.1. Glycaemic and Insulinemic Responses

The plasma glucose and insulin curves rose sharply and declined quickly after consumption of the control drink. The rises and declines in glucose and insulin were less steep when L-arabinose was added to the drink (Figure 1). The glucose peak was, on average, 1.1 mmol/L (15%) lower and the total glucose response (iAUC) was 32% lower when L-arabinose was added (Table 3). Glucose levels were significantly lower at 15 and 30 min, and L-arabinose seemed to prevent a drop in plasma glucose at 90 and 120 min (Figure 1a). Total insulin responses were 38% lower; the peak in insulin was 21 mU/L (52%) lower and occurred, on average, 12 min later (Table 3). A significantly lower insulin concentration was observed at 15, 30, and 45 min (Figure 1d).

#### 3.1.2. GLP-1 and GIP Responses

The control drink showed a sharp increase in plasma GLP-1, which declined quickly. This rise in GLP-1 was prolonged and the decline was delayed when L-arabinose was added to the drink (Figure 1g). The total GLP-1 response was 300% higher, the peak was 16 min later, and 6 pM (40%) higher (Table 3).

The plasma GIP curve rose immediately and declined after 90 min for the control drink (Figure 1j). When L-arabinose was added to the drink, the GIP response did not change from the baseline until 90 min (time effect = 0.99). From 15 to 90 min, the GIP response was significantly lower with L-arabinose compared to the control drink. L-arabinose addition reduced the peak (*p* < 0.001) for GIP with 88 pg/mL (43%) and prolonged the time to peak by 31 min (*p* < 0.001) (Table 3).

### 3.2. Fat Drink

#### 3.2.1. Glycaemic and Insulinemic Responses

Compared to the control drink, the fat drink showed a less sharp increase in plasma glucose, but the decline in glucose was almost as quick as the control drink (Figure 1b). Adding L-arabinose to the fat drink reduced the peak in plasma glucose by 0.5 mmol/L (8%). No effects of L-arabinose on the total response or the time of the peak were observed (Table 3).

The fat drink showed a similar insulin response compared to the control drink. Moreover, adding L-arabinose to the fat drinks resulted in a reduced total insulin response of 26%, and a 20 mU/L (45%) lower peak (Table 3). Moreover, insulin levels were significantly lower at 15, 30, and 45 min after consumption when L-arabinose was added (Figure 1e).

#### 3.2.2. GLP-1 and GIP Responses

The fat drink had a higher GLP-1 response compared to the control drink, which then decreased more slowly. Adding L-arabinose to the fat drink resulted in an increased overall GLP-1 response of 83% (Table 3). GLP-1 levels were higher at all time points from 30 to 120 min (Figure 1h).

The fat drink showed a higher GIP response than the control drink (Figure 1). Adding L-arabinose to the fat drink resulted in a 65% reduced total response, an on-average 18 min later peak, and an 88 pg/mL (48%) lower peak (Table 3). GIP levels were lower at all time points from 15 to 120 min (Figure 1k).

### 3.3. Starch Drink

#### 3.3.1. Glycemic and Insulinemic Responses

Overall, the starch drink had a more rapid and greater increase in glucose and insulin responses compared to the control and the fat drinks (Figure 1c,f). There were no statistical differences at individual time points in glucose or insulin levels between the starch drinks with or without L-arabinose. However, the glucose peak was lower (0.6 mmol/L; 7%) and the total glucose response was 32% lower. Moreover, the insulin peak was 25 mU/L (29%) lower and the overall insulin response was 33% lower (Table 3).

#### 3.3.2. GLP-1 and GIP Responses

The starch drink had a similar GLP-1 response compared to the control drink, declining more gradually (Figure 1i). L-arabinose addition to the starch drink led to a significant 53% increased GLP-1 response (*p* < 0.001) (Table 3).

Starch addition to the control drinks induced a higher and slower increase in GIP concentration compared to the control drink (Figure 1l). This was also visible in the 139 pg/mL (33%) lower peak (*p* < 0.001) (Table 3). L-arabinose addition to the starch drink reduced GIP immediately and significantly from 15 to 120 min postprandial.

### 3.4. Subjective Appetite Parameters, Thirst and Comfort

All subjective appetite parameters changed over time (all time effects *p* < 0.001) (see Appendix A), and no treatment×time effects were observed (all *p* > 0.05). No effects on thirst and comfort feelings were observed (all *p* > 0.05).

The subjects consumed on average 1019 ± 421 kcal (4.3 ± 1.8 MJ) from the lunch that was offered *ad libitum*. This intake was similar for all three test drinks and no effect of L-arabinose addition was observed (all *p*-values > 0.05) (see Appendix A).

### 3.5. Gastro-Intestinal Comfort

None of the gastrointestinal comfort markers—these being bloating, regurgitation, nausea, and flatulence—changed during the test morning after the consumption of the test products, and there were no differences between the test products whether L-arabinose was added or not (all *p*-values > 0.05) (see Appendix A).

## 4. Discussion

In the current study, the modifying effects of fat and starch on the efficacy of L-arabinose for lowering glycemic and insulinemic responses were assessed. It was confirmed that adding L-arabinose to sucrose-containing drinks lowers the glycemic and insulinemic responses in healthy subjects, by 15% and 52%, respectively. As hypothesised, adding fat or starch modified and reduced this effect. However, despite the effects being reduced, the glycemic peaks were still significantly lower than without the addition of L-arabinose. In the fat drink, L-arabinose reduced the glycemic peak by 8% and the insulin peak by 45%, while in the starch drink, the glucose peak was reduced by 7% and the insulin peak by 29%. Moreover, it was observed that, for all three model drinks, the addition of L-arabinose affected the post-prandial response of other physiological markers associated with glucose homeostasis, i.e., GLP-1 responses were increased and GIP responses were reduced. These findings suggest that L-arabinose affects postprandial glucose homeostasis in response to sucrose consumption, also when added to foods containing fat or starch.

To our knowledge, this is the first study that has assessed the effects of isolated food components on the functionality of L-arabinose in glucose homeostasis regulation. Although L-arabinose has been investigated often in simple single-nutrient (sucrose) liquid foods [10,11,12], only a few studies have investigated the effects of L-arabinose in more complex solid meals. For example, Halschou-Jensen investigated the effects of the addition of 5%, 10%, and 20% L-arabinose to sucrose- and/or starch-rich breakfast meals [15]. These meals were buns and muffins served with strawberry jam, buns with cheese, or liquified buns eaten as porridge. In this study, no effects were observed on glucose, insulin, or c-peptide concentrations. Our lab recently performed two studies on the addition of L-arabinose to ground cereals [14] and the replacement of sucrose with L-arabinose in muffins [13]. In both these studies, no effects were found on glucose peaks, but trends towards lower insulinemic responses were found (e.g., 10 mU/L lower peak after 8% L-arabinose replacement cereal [14]). However, these responses were not as clear and consistent as the effects in the simple liquid foods (fruit-flavoured drinks and water with added sucrose). Moreover, it was not clear which factor hindered the functionality of the L-arabinose, as the foods contained different nutrients; the texture, fat content, or starch content could have modified the functionality. Therefore, model drinks were used for the current study, carefully developed and only manipulated to be rich in one single nutrient, that being fat or starch. These model foods allowed for a better disentanglement of these effects.

Adding fat to the model drink resulted in a lower glucose peak of 1 mmol/L compared to the control drink with only sucrose. Yet, the addition of L-arabinose showed an additional effect on the plasma glucose peak—that is, 0.5 mmol/L (8%)—and on the insulin peak—that is, 20 mU/L (45%). It is known that the addition of fat lowers the gastric emptying rate and may therefore slow down the absorption of glucose [16,17]. These results suggest that even if L-arabinose and sucrose enter the duodenum at a slower pace, L-arabinose still binds to the sucrase enzyme and inhibits the absorption of glucose and fructose. The increase in GLP-1 and decrease in insulin response suggests that the GLP-1-producing L-cells in the proximal intestine detect the sucrose molecules despite L-arabinose being present.

Next to the modifying effect of fat, the modifying effect of starch was assessed. On top of the 50 g of sucrose which was present in the control drink, the starch drink contained an additional 50 g of available carbohydrates. Due to the higher level of available carbohydrates, it was hypothesised that the effect of L-arabinose could be overshadowed by the additional glucose from the starch. Indeed, the addition of starch led to a 0.8 mmol/L higher peak in glucose and also a 44 mU/L higher insulin peak after the consumption of the starch-rich drink compared to the control drink. However, when L-arabinose was added to the starch drink, there was still a 7% decrease in the glucose peak and a 29% lower insulin peak. So, L-arabinose was functional despite the additional glucose coming from starch in this model drink.

It has been speculated previously that L-arabinose not only inhibits sucrase but also inhibits maltase [11,15,21,22]. Maltase (α-glucosidase) is needed for the hydrolysis of maltose and is important for starch breakdown in the lumen of the duodenum. It is suggested that the inhibition of maltase by L-arabinose is not as pronounced as that of sucrase, but taking the amount of starch that is consumed every day in our diets, this may be an important additional strategy to lower blood glycemic responses [11,15]. Animal studies have shown inconsistent results in this respect. For example, Jurgonski did find a significant decrease in blood glucose when adding L-arabinose to starch challenges in rats [22], while Preuss did not find such effects [21]. In addition, Halschou-Jensen et al. did not find inhibiting effects when L-arabinose was added to solid mixed meals containing sucrose and starch from wheat flour [15]. From the current data, it is difficult to conclude whether L-arabinose also affected maltase, as it is impossible to disentangle the effects from the glucose coming from sucrose and the glucose coming from the starch present in the model drink. Moreover, for the present study, pregelatinised starch was chosen, as this starch is water-soluble at room temperature, which made it practically suitable for a model food and resembles the starch typically present in common foods such as bread and pasta. However, it cannot be ruled out that other factors such as physicochemical properties or the chain length of the starch molecules may affect the functionality of L-arabinose in starch digestion.

In order to better understand the underlying physiological response, not only postprandial glycemic and insulinemic responses were measured, but also GLP-1 and GIP responses. These are incretin hormones that are rapidly released after food consumption, and both stimulate insulin secretion and thus decrease glucose fluctuations after food intake [23]. GLP-1 responses were higher after all three drinks with added L-arabinose, ranging from 53% to 300%, while GIP responses were decreased by 33% to 48% and also showed later peaks. These higher GLP-1 responses and lower GIP responses are in line with earlier results [11,14]. The increase and prolonged rise in GLP-1 is presumably the result of higher and longer stimulation in the first part of the gut, while the lower GIP responses may be the result of a lower stimulation of k-cells in the duodenum and proximal jejunum. This is in line with the proposed mechanism of L-arabinose—that is, inhibiting the hydrolysis of sucrose in fructose and glucose, and thus the absorption of nutrients.

As for the simple sucrose-containing foods, the findings of earlier studies were reproduced [13,14]. Compared to previous studies, the current study used relatively high doses of L-arabinose: 5 g of L-arabinose per 50 g of sugar (10%). A sugar concentration of 10 g of sucrose/100 mL was chosen as this is equal to the amount of sucrose in sugar-sweetened drinks in the Netherlands [20]. A larger effect of L-arabinose may be accomplished by replacing sucrose with L-arabinose, which was demonstrated in one of the previous studies, where 30% of sucrose was replaced by L-arabinose and the glucose and insulin responses were even reduced by 22% and 67% [14]. These results show that L-arabinose can be used as a functional ingredient in sugar-rich drinks to prevent large fluctuations in glucose after the consumption of sugar-sweetened beverages. This may be especially relevant for (pre-)diabetic patients that are more prone to fluctuations in glucose.

None of the drinks showed consistent effects on subjective appetite responses or subsequent food intake, despite large effects on insulin and GLP-1. These findings are in line with other studies that did not see any effects on appetite or energy intake by L-arabinose [11,14,24].

The high doses of L-arabinose did not lead to any side effects or gastro-intestinal complaints, which is consistent with other studies [10,11,13,14]. If the hydrolysis of sucrose is inhibited, sucrose can flow from the small intestine into the colon, and be metabolised by microbiota, resulting in gastrointestinal symptoms such as flatulence and diarrhoea. Medicines such as acarbose, voglibose, and miglitol also inhibit digestive enzymes. These medicines are known for these side effects, due to the dumping of the undigested sugar fractions into the colon [25,26]. The current study shows that up to 1 g of L-arabinose per 10 g of available carbohydrates is a well-tolerated dose in liquids with a fast gastro-intestinal passage.

It is not clear to what extent L-arabinose is metabolised by the body. One of the previous studies has shown that L-arabinose is to some extent absorbed in the intestine, as up to 15% of supplemented L-arabinose was recovered in urine collected 24 h after consumption [14]. Moreover, it is likely that L-arabinose is also fermented by microbiota in the gut. This evidence comes from animal studies in pigs and rats [27,28], and, recently, in vitro and functional genomic studies showed that arabinose-containing fibres can be utilised by members of the Bacteroidetes phylum, which are abundantly present in the human colon [29,30]. However, more fundamental research is needed to further unravel the digestive and metabolic route of L-arabinose in the human body.

The advantage of adding L-arabinose to a sucrose-rich product is that the sweet taste is maintained and that other sensory properties, such as texture and mouthfeel, are not affected. The participants of the study were not able to tell the difference between the versions of the model drinks with and without L-arabinose. This study shows that even in model drinks rich in fat and starch, adding L-arabinose can affect glycemic control. The model drinks were manipulated beyond the nutrient content of existing foods, which makes them valid model drinks. The model drink rich in fat contained 3.8% *w*/*w* fat, which is a bit higher than full-fat milk in the Netherlands with 3.5% *w*/*w* fat [20]. The starch model drink contained 8.3% *w*/*w* pregelatinised starch and a total of 16.7% *w*/*w* available carbohydrates, which is twice as high as commercially available vanilla custard, which contains 4.6% *w*/*w* starch and 10.0% *w*/*w* total available carbohydrates [20]. Hence, this study illustrates the opportunities and bandwidth of foods to which L-arabinose can be added. It needs to be further investigated what the other food matrix related factors are that affect the functionality of L-arabinose.

Lastly, as the study population consisted of healthy, lean, young individuals, their glucose levels were relatively well-controlled. The range of foods and their properties, such as nutrient content and matrix, that may be supplemented with L-arabinose need to be investigated further in populations with impaired glucose tolerance and/or diabetes. This study illustrates how novel functional ingredients can be derived from plant food processing side-streams.

## 5. Conclusions

In conclusion, this study shows that L-arabinose addition to sugar-sweetened drinks, even with other nutrients present, such as fat or starch, flattens the blood glucose, insulin, and GIP response curves and increases GLP-1 concentrations in circulation in healthy subjects. L-arabinose addition to sugar-sweetened drinks may be beneficial for people that want to dampen their glycemic and insulinemic responses, such as people with (or at risk of) type 2 diabetes. Moreover, their glucose tolerance may be improved by reducing GIP secretion and increasing GLP-1 secretion. The advantage of adding L-arabinose to a sucrose-rich product is that the sweet taste is maintained. Therefore, L-arabinose addition to real sugar-sweetened foods with some starch or fat seems to be a promising strategy to affect acute postprandial glycemic responses. This may be a strategy to target consumers that are keen on drinks or foods that are high in sucrose content and do not want to use other (artificial) sweeteners. Further research should further focus on applications in solid foods and patient populations that would benefit from better glucose control, such as (pre)diabetics.

## Figures and Tables

**Figure 1 foods-11-00157-f001:**
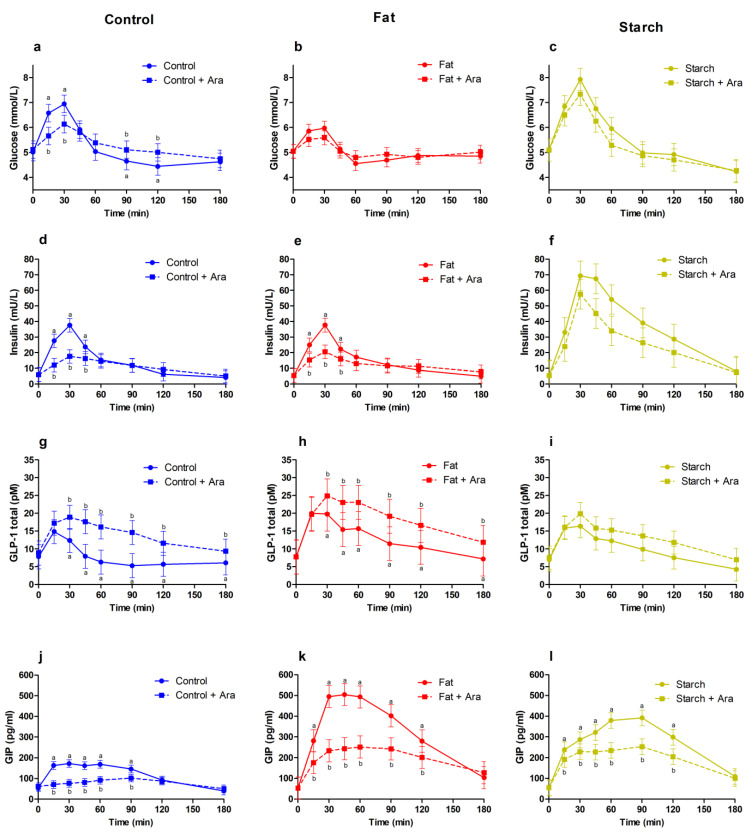
Glycemic and hormone concentrations (mean and SEM) before and after the consumption of each of the 6 drinks: control drink = blue, control with added fat = red, control with added starch = green. Dotted lines represent drinks with added arabinose, solid lines represent drinks without arabinose. (**a**–**c**) Glucose curves, (**d**–**f**) insulin curves, (**g**–**i**) GLP-1, (**j**–**l**) GIP. Different letters represent statistical differences (*p* < 0.05) between treatments at a specific time point (mixed-model ANOVA, post hoc contrasts with Tukey correction).

**Table 1 foods-11-00157-t001:** Baseline characteristics of the study subjects (mean ± SD).

	Total (*n* = 23)	Men (*n* = 12)	Women (*n* = 11)
Age (y)	23.8 ± 3.3	24.4 ± 3.3	23.2 ± 3.3
Body weight (kg)	74.5 ± 11.2	77.6 ± 9.9	71.2 ± 12.6
Height (m)	1.79 ± 0.09	1.84 ± 0.08	1.74 ± 0.07
BMI (kg/m^2^)	23.1 ± 2.7	22.8 ± 1.6	23.5 ± 3.7

**Table 2 foods-11-00157-t002:** Nutrient content, energy content, weight, and liking of the test products.

	Water	Sucrose	L-Arabinose	Fat	Starch	Energy	Weight	Liking ^1^
	(g)	(g)	(g)	(g)	(g)	(kcal)	(g)	
Control	500	50	0	0	0	200	550	5.2 ± 1.2 ^a^
Control + Ara	500	50	5	0	0	200	555	5.0 ± 1.3 ^ad^
Fat	500	50	0	22	0	398	572	4.2 ± 1.0 ^bd^
Fat + Ara	500	50	5	22	0	398	577	4.1 ± 1.3 ^b^
Starch	500	50	0	0	50	400	600	3.1 ± 1.4 ^c^
Starch + Ara	500	50	5	0	50	400	605	2.7 ± 1.3 ^c^

^1^ Liking was measured on a 7-point Likert scale, anchored from “not liked at all” to “very much liked”. Mean ± SD are presented. Different letters indicate significant differences in liking between test products (*p* < 0.05). Nutrient content was estimated based on the ingredients and energy content was calculated by using the Atwater factors—that is, 4 kcal for carbohydrates and 9 kcal for fat. The energetic value of L-arabinose was ignored, as its metabolic energetic content is unknown.

**Table 3 foods-11-00157-t003:** Postprandial responses (iAUC_0–180min_), peak (C_max_) and T_peak_ (time to peak) of glucose, insulin, GLP-1 and GIP after a single dose of the sucrose solution without or with L-arabinose and without (control) or with fat or starch.

		Without L-Arabinose	5 g L-Arabinose	*p*-Value ^2^
**Glucose**				
Peak (C_max_) (mmol/L)	Control	7.4 [7.0; 7.7] ^1^	6.3 [5.9; 6.6]	<0.001
	Fat	6.4 [6.1; 6.6]	5.9 [5.6; 6.1]	<0.01
	Starch	8.2 [7.7; 8.6]	7.6 [7.1; 8.0]	<0.05
Time to peak (T_peak_) (min)	Control	27 [22; 32]	33 [28; 38]	NS
	Fat	28 [17; 39]	32 [21; 43]	NS
	Starch	34 [27; 41]	29 [22; 36]	NS
Response (AUC) ^3^ (mmol/Lxmin)	Control	91.7 [61.2; 122.3]	61.9 [31.4; 92.5]	<0.01
	Fat	46.8 [29.3; 64.3]	41.4 [24.0; 58.9]	NS
	Starch	151.1 [104.8; 197.3]	103.5 [57.2; 149.7]	<0.05
**Insulin**				
Peak (C_max_) (mU/L)	Control	41.0 [33.4; 48.6]	19.7 [12.1; 27.3]	<0.001
	Fat	44.0 [35.6; 52.4]	24.1 [15.7; 32.4]	<0.001
	Starch	85.1 [69.1; 101.1]	60.6 [44.6; 76.6]	<0.01
Time to peak (T_peak_) (min)	Control	32 [23; 40]	44 [36; 53]	<0.01
	Fat	31 [20; 41]	38 [27; 48]	NS
	Starch	50 [39; 60]	42 [32; 52]	NS
Response (AUC) (mU/Lxmin)	Control	1538 [1124; 1952]	952 [538; 1366]	<0.001
	Fat	1654 [1321; 1987}	1228 [896; 1561]	<0.01
	Starch	5501 [4358; 6644]	3662 [2519; 4805]	<0.01
**GLP-1**				
Peak (C_max_) (pM)	Control	15.4 [10.4; 20.5]	21.7 [16.7; 26.6]	<0.001
	Fat	22.3 [15.4; 29.2]	27.5 [20.6; 34.4]	NS
	Starch	19.0 [14.1; 23.8]	22.1 [17.3; 26.9]	NS
Time to peak (T_peak_) (min)	Control	24 [9; 39]	40 [25; 55]	<0.001
	Fat	30 [13; 47]	52 [35; 69]	0.05
	Starch	38 [22; 54]	47 [31; 63]	NS
Response (AUC) (pM × min)	Control	268 [45; 491]	803 [581; 1026]	<0.001
	Fat	987 [394; 1581]	1805 [1212; 2399]	<0.01
	Starch	627 [418; 836]	962 [752; 1171]	<0.001
**GIP**				
Peak (C_max_) (pg/mL)	Control	204 [179; 230]	116 [90; 142]	<0.001
	Fat	578 [507; 649]	278 [207; 348]	<0.001
	Starch	421 [370; 472]	282 [231; 333]	<0.001
Time to peak (T_peak_) (min)	Control	42 [27; 57]	73 [58; 88]	<0.001
	Fat	45 [33; 57]	63 [51; 76]	<0.05
	Starch	83 [71; 96]	77 [65; 89]	NS
Response (AUC) (pg/mL × min)	Control	6375 [3985; 8765]	5161 [2771; 7551]	NS
	Fat	35,750 [28,255; 43,244]	23,121 [15,627; 30,616]	<0.01
	Starch	28,383 [22,081; 34,684]	21,755 [15,453; 28,057]	NS

^1^ Least square means (LSMEANS) with 95% confidence intervals between brackets. ^2^
*p*-value mixed model ANOVA contrast between without and with arabinose addition. *p*-values were categorised into <0.001, <0.01, <0.05, and NS (non-significant). ^3^ AUC = Area Under the Curve.

## Data Availability

The data presented in this study are available on request from the corresponding author. The data are not publicly available due to privacy and ethical reasons.

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
