# Peer review of "Efficacy of L-Arabinose in Lowering Glycemic and Insulinemic Responses: The Modifying Effect of Starch and Fat"

_foods, 2022, doi:10.3390/foods11020157_

Round 1

Reviewer 1 Report

This paper describes the effect of fat or starch on the efficacy of L-arabinose in lowering glycemic and insulinemic responses. This well-written paper potentially attracts the interest of readers in food science. I would recommend the minor points listed below.

  • Space is needed between number and unit, for example, 50g in line 14.
  • The words “Error! Reference source not found” should be removed in lines 95, 129, and 142.

I hope these comments will be helpful.

Author Response

Thank you for your suggestions. We added spaces between numbers and units throughout the document and replaced the error messages for the table numbers (these were internal references).

Reviewer 2 Report

The paper “Efficacy of L-arabinose in lowering glycemic and insulinemic 2 responses: the modifying effect of starch and fat” presents the the effect of fat or starch in a sugary drink on the efficacy of L-arabinoseThe text and results presented in the paper have copious amount of flaws. So the manuscript is unacceptable in its present form

Material and methods section missing appropriate references in each subsection. Please mention appropriate reference

Describe the methods used for Nutrient content, energy content, weight and liking of the test products in detail in material and method section with the makes and model of the instruments used so that the readers can have proper knowledge

Line 130 Reference source not found.. Double full stop have been used. Please make correction

Avoid using we have not seen the effect or our results. Use the words such as in this study.

The results should be compared with recent studies properly

Many references are older than 15 years. Please use recent studies.

Some references should be added from the journal foods mdpi.

There are some English and grammatical mistakes please check carefully.

Author Response

Reviewer 2:

The paper “Efficacy of L-arabinose in lowering glycemic and insulinemic responses: the modifying effect of starch and fat” presents the effect of fat or starch in a sugary drink on the efficacy of L-arabinose. The text and results presented in the paper have a copious amount of flaws. So the manuscript is unacceptable in its present form.

Answer: Thank you for your critical review. We hope to improve the paper by responding to the suggestions raised. We will address the points raised by the reviewer one by one.

  1. Material and methods section missing appropriate references in each subsection. Please mention the appropriate reference. Describe the methods used for Nutrient content, energy content, weight, and liking of the test products in detail in the material and method section with the makes and model of the instruments used so that the readers can have the proper knowledge.

Answer:

To our knowledge, the requested details are there, but we tried to add more details. The methods used for the nutrient content analyses have been added in the footnote of the table: “Nutrient content was estimated based on the ingredients, energy content was calculated by using the Atwater factors, that is 4 kcal for carbohydrates and 9 kcal for fat. The energetic value of L-arabinose was ignored, as its metabolic energetic content is unknown.” Methods of liking measurements were already described in lines 139-140 and in the footnote of Table 2, we now also added the anchors under the table and referred in line 188 to the section 2.3 test products. Details of materials to reproduce the products have already been mentioned in section 2.3.1. Details of the biochemical measurements are already been mentioned in section 2.5.

  1. Line 130 Reference source not found. Double full stops has been used. Please make correction

Answer: This has been corrected, this was an internal reference to the table.

  1. Avoid using we have not seen the effect or our results. Use the words such as in this study.

Answer: We assume that he or she means we use “we” and “our” too often. We have revised the manuscript and now use “we” and “our” only if deemed necessary.

  1. The results should be compared with recent studies properly.

Answer: The results of our experiment are compared with other experiments in several paragraphs of the discussion. For example, they are compared with other studies with complex foods in lines 364-382, and with other studies with liquids in lines 432-439.

  1. Many references are older than 15 years. Please use recent studies.

Answer: The references used are the only ones that are available on the topics that we discuss. We do not see a reason to not use references older than 15 years in this context as there is no newer data available.

  1. Some references should be added from the journal foods mdpi.

Answer: We do not agree with this remark, there is no reason that including references from the MDPI journals will improve the academic quality of the paper.

  1. There are some English and grammatical mistakes please check carefully.

Answer: We reread the paper carefully and checked for English and grammatical mistakes.

Reviewer 3 Report

This manuscript is quite interesting crossover designed human study to deal with the effect of fat or starch in a sugary drink on the efficacy of L-arabinose. I think it can be minor revision, as it is the revision of some typos (e.g. line 129-130)

Author Response

Thank you for your interest in the study. We have looked at the referred section and checked the document on typos.

Reviewer 4 Report

The manuscript is written but similar results has been published before and it has no novelty for the readers.  

Please check the manuscript “The effect of replacing sucrose with L-arabinose in drinks and cereal foods on blood glucose and plasma insulin responses in healthy adults”

https://www.sciencedirect.com/science/article/pii/S1756464620303388

Author Response

Thank you for referring us to this paper. We are fully aware of that specific paper (it is our own) and we also refer to it several times in the introduction and discussion. We do not agree that there is no novelty for the reader. The novelty of the current experiment is explained in lines 44 to 62.

In short, experiments have shown that adding L-arabinose to simple drinks or gels containing sucrose reduces postprandial glycemic responses. The novelty of the current manuscript is that it describes the results of a model food experiment that explored the modifying effect of isolated food components (starch and fat) on the efficacy of L-arabinose in sucrose-rich foods. No such studies have been performed before. Studies with real-life foods show inconsistent results as they may differ in many ways from simple drinks or gels. Which also holds for the cereals in the paper to which the reviewer refers. Therefore, it is essential that the impact of other food components should be studied systematically with model food studies before L-arabinose can be implemented into real-life and complex foods.

We think that it is specifically of interest for the readers of the journal “Foods” as it shows how food processing or the addition of other ingredients/nutrients can alter the biological functionality of compounds. It, therefore, emphasizes that it is critical that the biological functionality is tested within the food application before any health related claims can be made.

Reviewer 5 Report

The authors evaluated the modifying effects of starch and fat upon the efficacy of L-arabinose to lower glycemic and insulinemic responses. They observed lowering of these responses after the addition of L-arabinose to the studied subjects' drinks  .They found that L-arabinose affects postprandial glucose homeostasis in response to sucrose consumption also when added to foods having starch or fat.

It is well written paper and the subject should be of interest to scientists and the food and pharmaceutical industries. The experimental part is well designed and approved by the ethical committee.The bibliography is up to date..

Please correct and explain at line 95 the remark under parenthesis (Error!Reference source not found)

My suggestion is  minor corrections.

Author Response

Thank you for your interest in the paper. We have adjusted the reference link (which was an internal link to a table).

Round 2

Reviewer 2 Report

All the comments are done well. The paper is now good for publication.

Reviewer 4 Report

The current experiment should be the part of previous. 

I do not agree with the novelty  statement provided by the authors to create complete new manuscript.